# Optimal Tasking of Ground-Based Sensors for Space Situational Awareness Using Deep Reinforcement Learning

**DOI:** 10.3390/s22207847

**Published:** 2022-10-16

**Authors:** Peng Mun Siew, Richard Linares

**Affiliations:** Department of Aeronautics and Astronautics, Massachusetts Institute of Technology, Cambridge, MA 02139, USA

**Keywords:** space situational awareness, sensor tasking, deep reinforcement learning

## Abstract

Space situational awareness (SSA) is becoming increasingly challenging with the proliferation of resident space objects (RSOs), ranging from CubeSats to mega-constellations. Sensors within the United States Space Surveillance Network are tasked to repeatedly detect, characterize, and track these RSOs to retain custody and estimate their attitude. The majority of these sensors consist of ground-based sensors with a narrow field of view and must be slew at a finite rate from one RSO to another during observations. This results in a complex combinatorial problem that poses a major obstacle to the SSA sensor tasking problem. In this work, we successfully applied deep reinforcement learning (DRL) to overcome the curse of dimensionality and optimally task a ground-based sensor. We trained several DRL agents using proximal policy optimization and population-based training in a simulated SSA environment. The DRL agents outperformed myopic policies in both objective metrics of RSOs’ state uncertainties and the number of unique RSOs observed over a 90-min observation window. The agents’ robustness to changes in RSO orbital regimes, observation window length, observer’s location, and sensor properties are also examined. The robustness of the DRL agents allows them to be applied to any arbitrary locations and scenarios.

## 1. Introduction

The number of resident space objects (RSOs) in Earth’s orbit has been steadily increasing in recent decades and is driven by the increasing number and scale of commercial satellite constellations in low Earth orbit (LEO). Space situational awareness (SSA) or space domain awareness is the ability to detect, identify, catalog, and track the RSOs currently orbiting the Earth. SSA plays an essential role in ensuring the safe operation of satellites and space sustainability, as the RSOs can pose a collision hazard to crucial space assets and future space launches. The United States Space Surveillance Network (SSN) maintains one of the most extensive databases of RSOs using a collection of more than 30 globally distributed ground-based and space-based sensors [1]. These sensors are required to retain custody of RSOs that are a few orders of magnitude larger in number. There are currently more than 47,000 pieces of space debris being tracked by the United States SSN, of which 6500 are active payloads and 19,500 are debris that is being regularly tracked (Numbers taken from https://www.space-track.org/ accessed 1 July 2022).

The U.S. SSN consists mainly of ground-based sensors, and some of these sensors have a limited field of view and are required to cover a significantly larger field of regard. Each sensor has significantly different capabilities and limitations, such as sensor slew time, sensor resolution, and mission priorities. Ground-based optical telescopes have a limited observation window where they are only operational during nighttime, making it infeasible to observe all RSOs in the limited observation window. The current U.S. SSN can reliably track RSOs that are larger than 10 cm in LEO, and according to the European Space Agency Meteoroid and Space debris Terrestrial Environment Reference (MASTER-8) model [2], there are currently over a million RSOs that are within 1 cm to 10 cm that is too small to be tracked by the U.S. SSN but can still pose a lethal collision threat to satellites.

With the increased space activities and the advancement in space object sensing capabilities, the number of RSOs in the U.S. SSN catalog is expected to increase further. Therefore, the sensors in the U.S. SSN need to be efficiently tasked to ensure custody of these RSOs and provide early warnings of potential collision hazards while considering each sensor’s constraints and limitations. Traditionally, the U.S. SSN uses a daily centralized tasking algorithm with a decentralized scheduler [3]. The RSOs within the catalog are sorted by the centralized tasker into five categories depending on their priority, followed by a subcategory (suffix) to indicate the number of tracks per day and how many observations each track. Each SSN sensor will then schedule observations based on the daily tasking list provided by the centralized tasker while considering the sensor specifications and limitations. The SSN sensor scheduler will first dedicate sensing resources to the high-priority RSOs and defer the lower-priority RSOs to future passes when needed.

The SSA sensor tasking problem can be formulated as a classical resource allocation problem, where the goal is to maximize the return while constrained by limited resources. The long-term sensor tasking problem for SSA is inherently hard as it suffers from the curse of dimensionality, for which the complexity of the problem increases exponentially with the number of targets and observation window. This is further complicated by the nonlinearity of the SSA sensor tasking problem. Classical sensor tasking algorithms for SSA tend to be myopic, where only the short-term benefits of each action are considered. Erwin et al. (2010) looked into an information-theoretic approach using a Fisher information gain (FIG) based sensor tasking algorithm where the objective is to maximize the potential reduction of the covariance matrix at each observation step [4]. This work was later extended by Williams et al. to incorporate stability criteria via a Lyapunov exponents-based sensor tasking algorithm [5]. Williams et al. pointed out that the naive FIG-based sensor tasking algorithm can result in little to no observations of objects with low values of FIG with respect to the other RSOs and possibly lead to divergence in its uncertainty. Gehly et al. (2018) proposed the usage of the Rényi divergence as an information gain functional for the SSA sensor tasking algorithm [6]. The Rényi-based tasking scheme favors measurements that produce the greatest relative change in covariance, regardless of the magnitude of the initial covariance.

Several researches have been conducted on solving the long-term sensor tasking problem. Sunberg et al. (2015) uses a dynamic programming approach where a receding horizon solution is used [7]. Fedeler and Holzinger (2020) proposed a Monte Carlo-based tree search algorithm to approximately solve the long-term sensor planning problem [8]. Their proposed partially-observable Monte Carlo hybrid update planning (POMCHUP) algorithm uses a tree search algorithm to maximize information gain over a receding horizon of observation. However, substantial computational power is still required to evaluate the possible actions. Instead of using an information-theoretic approach, Frueh et al. (2018) model the SSA sensor tasking problem as an optimization problem and approximate the sensor tasking problem with a near-optimal formulation and a greedy solution mechanism [9].

Moreover, most SSA sensor tasking algorithms are formulated based on specific operator-defined metrics; such as observing the greatest number of unique RSOs, keeping the overall RSO uncertainties within an upper uncertainty bound, maximizing sky coverage, maximizing Fisher information gain, minimizing the mean time between observations, and operator assigned RSO viewing priority. Often time, these objective contradicts one another. For example, observing the most number of unique RSOs would require minimizing action slew time and observing RSOs that are in close proximity regardless of their uncertainties. There is little research done on developing a general sensor tasking formulation that can adapt to a diverse range of operator objectives.

Deep reinforcement learning (DRL) have demonstrated groundbreaking performance over a wide range of domain and have even outperformed human experts in complex tasks that require long-term planning [10,11,12]. For example, silver et al. (2016) successfully trained a DRL agent to play GO, where the trained agent could defeat a professional GO player [10]. Due to its large action space, GO has a vast number of possible board positions in the order of $10^{170}$. The DRL agent was able to overcome the curse of dimensionality and learn high-level abstractions of the problem for long-term decision-making. In [11,12], DRL agents were successfully trained to play complex computer games involving a long-time horizon, imperfect knowledge, and complex, continuous, continuous state-action spaces. The application of DRL techniques for SSA was first studied by Linares and Furfaro (2016), where an actor-critic-based DRL agent was trained to track a small population of near Geosynchronous (GEO) RSOs [13]. Linares and Furfaro (2017) then extended their previous work and demonstrated the usage of an Asynchronous Advantage Actor-Critic (A3C) DRL agent to track a larger population of 100 and 300 near GEO RSOs, respectively, [14]. However, in both studies, it was assumed that all RSOs were always observable, i.e., there exist ground-based sensors all over to world to ensure complete coverage of all RSOs at all times. Besides, their scenarios do not consider the action slew rate and use a constant propagation time of 30 s between observations regardless of the required slew angle. Little and Frueh (2020) successfully applied two different DRL algorithms—ant colony and distributed Q Learning—to the SSA problem [15]. Both DRL algorithms outperform the traditional greedy algorithm and the Weapon-Target Assignment algorithm in tracking the most number of RSOs. However, their proposed solutions are specific to a given observation window and RSO catalog, and the DRL agents must be retrained for each scenario and RSO catalog. Part of this work was previously presented at the 2021 AAS/AIAA Astrodynamics Specialist Conference, where an actor-critic DRL agent was successfully applied to task a single ground-based optical sensor with finite action slew rate to observe near GEO RSOs [16]. Roberts et al. (2021) used a similar framework to [16] to solve the sensor tasking problem for a space-based system [17].

In this paper, we focus on the catalog maintenance subproblem of the SSA task. This problem involves optimally tasking a single ground-based optical telescope to obtain new measurements for RSOs with known orbits. These new measurements serve to maintain custody and improve orbit estimates of objects within the SSA catalog. Four actor-critic DRL agents with different neural network architectures are studied in this work. DRL agents’ high-level, low-dimensional abstraction capabilities are leveraged to train an agent that can generalize across different resident space object populations, different environment setups and initialization, and a range of operator objectives. Two different myopic policies are used as baseline comparisons to evaluate the performance of the DRL agent. The training and evaluation of the DRL agents are conducted using an in-house SSA environment. The SSA environment uses the Simplified General Perturbation 4 (SGP4) propagation model with no additional external perturbations to propagate the RSOs. Meanwhile, the RSOs covariance is propagated and updated using an Unscented Kalman Filter (UKF) formulation. The performance of the various policies is evaluated in terms of final mean trace covariance and the cumulative number of unique RSOs observed at the end of the 90-min observation window.

The main contributions of this work are:A flexible SSA simulation environment is developed to train and evaluate deep reinforcement learning agents.The SSA environment can support arbitrary sensor location, RSOs population and covariance, observation window, and sensor properties (action slew time, settle time, dwell time, measurement model, measurement noise, and process noise).Four different actor-critic DRL agents with various neural network architectures were trained to solve the SSA sensor tasking problem for a single ground-based optical sensor.Monte Carlo runs are used to benchmark the performance of the DRL agents against myopic policies, and the robustness of the DRL agents to variation in the RSO orbital regime, observation window, observer location, and sensor properties are also analyzed.

The paper is organized as follows. First, the development of the SSA environment is presented in Section 2. Second, an overview of the DRL problem formulation and the DRL agent architectures studied in this paper are presented in Section 3. Third, the performance and robustness analysis of the various DRL agents are presented in Section 4. Finally, the paper is concluded and suggestions for future work are provided in Section 5.

## 2. Space Situational Awareness Environment

The custom SSA environment used for the training of DRL agents and performance evaluation is developed in this section. The SSA environment uses the OpenAI Gym framework [18] and is responsible for keeping track of the RSOs’ states and covariance, creating the observation input for the various action policies, generating noisy measurement for each RSOs within the sensor’s field of view, and computing the immediate reward of the current action. Observation refers to the state of the environment that is accessible to the agent or action policy during the decision-making (sensor tasking) process. In contrast, measurement refers to the RSO’s Cartesian position reading generated from successfully detecting an RSO. The SSA environment consists of a single ground-based narrow field of view optical telescope with a finite action slew rate, settle time, dwell time, and data readout time. The RSO states and covariance are propagated and updated using a UKF formulation. The SSA environment supports arbitrary sensor location, RSO population, number of RSOs, observation window, and user-configurable sensor parameters, such as action slew time, action settle time, action dwell time, measurement noise, and process noise.

The sensor properties are modeled based on the Zimmerwald SMall Aperture Robotic Telescope (ZimSMART) operated by the Astronomical Institute of the University of Bern, Switzerland [19]. The ZimSMART telescope is assumed to be equipped with an additional Light Detection and Ranging (LiDAR) system to produce Cartesian coordinate measurements for all RSOs within the current field of view, and two separate motors are used for azimuth and elevation control of the telescope mount. Based on data provided in [19], the readout time is approximated as 7.7 s, while the slew and settle time is approximated such that it takes a total of 4.55 s for every $4^{{}^{\circ}}$. A constant dwell (exposure) time of 1.3 s is used. The optical sensor parameters are shown in Table 1. The propagation time for a single time step can vary between 9 s to 209.2 s depending on the slew angle between the current pointing direction and the selected action.

The ground-based optical sensor can point to any arbitrary pointing direction above the minimum viewing horizon, resulting in a highly complex and intractable problem with infinitely many pointing directions within the field of regard. The sensor tasking problem is simplified by discretizing the continuous action and observation space into a finite number of discrete actions and observations, where the sensor’s field of view is used to discretize the field of regard into non-overlapping patches of the sky. The field of regard spans the full azimuth angles ranging from 0${}^{{}^{\circ}}$ to 360${}^{{}^{\circ}}$, whereas the minimum viewing horizon constrains the elevation angles. For this study, a conservative minimum viewing horizon of 14${}^{{}^{\circ}}$ is chosen, where the sensor cannot observe any objects with an elevation angle lower than 14${}^{{}^{\circ}}$ and this limits the sensor’s elevation angles to between 14${}^{{}^{\circ}}$ and 90${}^{{}^{\circ}}$. With a sensor’s field of view of 4${}^{{}^{\circ}}$× 4${}^{{}^{\circ}}$, this results in 19 elevation bins and 90 azimuth bins, for a combination of 1710 possible patches of the sky that the sensor can point toward, i.e., a finite action space with 1710 possible pointing directions.

Figure 1 shows the process flow of the SSA environment. The SSA environment consists of six modules, and their respective outputs are listed below and indicated by a red arrow. During the initialization of the SSA environment, the initial orbital elements $x_{0}$ of the RSOs and their associated covariance $P_{0}$ are initialized.
(1)$$x={\left[e,\;i,\;\Omega ,\;\omega ,\;n,\;M_{0}\right]}^{T}$$
where *e*, *i*, Ω, ω, *n*, and $M_{0}$ are the eccentricity, inclination, right ascension of ascending node, argument of perigee, mean motion, and mean anomaly of the RSO.

The observation generation module then generates a grid-based observation array that functions as the observation input for the various action policies. The action policy module uses the incomplete information in the observation grid to select the next pointing direction. The action policy module is user configurable and can be swapped out to any arbitrary action policy or RL agent. The required slew motion is used to compute the total action time, and the RSOs are then propagated forward in time using a UKF formulation with an SGP4 propagation model. After propagating the RSOs, any RSOs within the sensor’s current field of view are then identified, and a noisy measurement is generated for these RSOs. It is assumed that the sensor can perfectly assign each measurement to the correct RSO. A UKF formulation is then used to update the orbital elements and covariance of the observed RSOs. The SSA environment then generates a new grid-based observation array, and the whole process is repeated until a termination criterion is reached. The rollout terminates when one of the following conditions are met:The reward score falls below a particular threshold.There are no RSOs in the field of regard.The end of the observation window.

### 2.1. Environment Initialization

Successful training of DRL agents typically requires a large collection of training data with sufficient variation to cover all possible scenarios. The variation in the training data also encourages robustness and prevents the DRL agent from over-fitting to a particular subset of scenarios. Therefore, instead of initializing our environment using Two Line Element (TLE) sets, the environment is randomly initialized to ensure a good diversity among the RSO population used for training and performance evaluation. However, naively sampling over all possible orbital elements would result in a uniformly distributed RSO population across the globe, where most of the randomly generated RSOs do not enter the sensor’s field of regard during the 90-min observation window. Instead, kernel density estimation (KDE) sampling is used to randomly initialize the orbital elements of the RSO population at each training and evaluation episode. The KDE sampling maximizes training efficiency by ensuring that a large percentage of the randomly sampled RSOs enters the sensor’s field of regard during each training episode. A KDE distribution is generated by first identifying the orbital elements of RSOs within the sensor’s field of regard during the different points of the observation window. The KDE algorithm is then used to fit the underlying orbital elements of these RSOs into a Gaussian Mixture distribution. Figure 2a shows the evolution of the instantaneous number of RSOs within the field of regard at different propagation times for an environment initialized with 100 randomly sampled RSOs, while Figure 2b shows the cumulative number of unique RSOs that have entered the field of regard since the beginning of the observation window. The KDE sampling initialization method resulted in more than triple the number of observable RSOs within the field of regard compared to the naive uniform sampling initialization method throughout the observation window. Furthermore, more than 65% of the uniformly sampled RSOs never enter the field of regard compared to less than 5% when the RSOs are initialized using the KDE sampling method. However, the KDE sampling method is an observer location and observation window-dependent initialization method, and a new KDE model needs to be generated whenever the observer’s location or the observation window is changed.

The RSOs are assumed to be well-tracked, where their uncertainties can be initialized as a Gaussian distribution with a diagonal covariance matrix. The first and third quartile of 47 TLE associated uncertainties data from Osweiler (2006) [20] is used to guide the sampling range for diagonal covariance matrix and is presented in Table 2. The upper and lower sampling bound for the mean motion is increased by a factor of 100 to inflate the randomly initialized RSOs’ uncertainties artificially.

### 2.2. Observation Generation

After discretization of the field of regard by the sensor’s field of view, the observation array takes the form of a three-dimensional array with the dimensions: 19 by 90 by x, where the first two dimensions represent all possible azimuth and elevation pointing directions and x is the number of information layers. The discretized observation space allows the SSA environment to accommodate RSO catalogs with any arbitrary number of RSOs. Eleven information layers consisting of RSO state and uncertainties information, as shown in Table 3 represent the current state of the partially observable environment. When multiple RSOs are present in the same discretized grid, data of the RSO with the largest uncertainties are used to populate the observation array.

The observation array uses an azimuth-invariant formulation, populated such that the current pointing direction is always at the center of the observation array, i.e., the 45th row of the observation array. The remaining rows are then populated based on their relative azimuth angle from the current pointing direction. The low slew rate results in a large range of action slew time, where the sensor can take anywhere between 9 s to 209.2 s to slew from one pointing direction to another and generate a measurement. The required slew time to reach the respective pointing directions needs to be considered when formulating the observation array to better reflect the true state of the environment at the end of the slewing motion. Thus, the observation grid is partitioned into seven regions, and each region uses a different propagation time for data population depending on the region’s approximate action time (sum of slew time, settle time, and dwell time), as shown in Figure 3. The white cell indicates the current pointing direction, and the seven regions are color-coded with a different color. The propagation time steps of 15 s, 45 s, 75 s, 105 s, 135 s, 165 s, and 195 s are used for the seven regions as we move outward from the current pointing direction. This partitioning of the observation grid helps keep the computational time low while ensuring that it reflects the true expected environment state at the end of the slew motion.

### 2.3. Action Policy

The action policy module is responsible for selecting the next pointing direction based on incomplete information within the observation grid. The same discretization in the observation generation module is used to discretize the action space, where the continuous action space is discretized into 19 discrete elevation pointing angles and 90 discrete azimuth pointing angles. This results in a combination of 1710 discrete actions. The action policy module is user configurable and can be swapped out to any arbitrary action policy or DRL agent. Details of the various DRL agents studied in this work are presented in Section 3, while implementation details of the myopic policies are introduced in Section 4. The myopic policies serve as baseline comparisons to benchmark the performance of the DRL agent.

### 2.4. Kalman Filter Propagation

The required action time to slew from the current pointing direction to the new action and the time required to generate a measurement (settle and dwell time) is then calculated and used to propagate the RSOs. The RSOs are propagated using an SGP4 model with no additional external perturbation, and no station-keeping maneuvers are assumed. Meanwhile, the RSO covariance is propagated using a Kalman Filter formulation as shown in Equations (Equation 2) and (Equation 3).
(2)$$x_{k|k-1}=SGP4\left(x_{k-1|k-1},\;\delta t\right)$$
(3)$$P_{k|k-1}=FP_{k-1|k-1}F^{T}+Q$$
where *F* and *Q* are the simplified state transition model (assuming no external perturbations) and the process noise and are given by Equations (Equation 4) and (Equation 5), respectively. The process noise *Q* is selected based on values used in past literature [21].
(4)$$\begin{array}{cc} F & =SGP4(\;\cdot \;,\;\delta t)\\ \; & \approx \left[\begin{array}{cccccc} 1 & 0 & 0 & 0 & 0 & 0\\ 0 & 1 & 0 & 0 & 0 & 0\\ 0 & 0 & 1 & 0 & 0 & 0\\ 0 & 0 & 0 & 1 & 0 & 0\\ 0 & 0 & 0 & 0 & 1 & 0\\ 0 & 0 & 0 & 0 & \delta t(\frac{360}{86400}) & 1 \end{array}\right] \end{array}$$
where δt is the action time.
(5)$$Q=\left[\begin{array}{cccccc} 1.2736\times 10^{-14} & 0 & 0 & 0 & 0 & 0\\ 0 & 1.9226\times 10^{-14} & 0 & 0 & 0 & 0\\ 0 & 0 & 1.6388\times 10^{-17} & 0 & 0 & 0\\ 0 & 0 & 0 & 2.6206\times 10^{-10} & 0 & 0\\ 0 & 0 & 0 & 0 & 2.3751\times 10^{-10} & 0\\ 0 & 0 & 0 & 0 & 0 & 7.7148\times 10^{-17} \end{array}\right]$$

### 2.5. Measurement Generation

At the end of the propagation, observability conditions for all RSOs are calculated where any RSOs located within the observer’s current field of view are identified. However, the sensor’s effective azimuth viewing angle projected onto the discretized field of regard varies depending on the elevation angle, where it increases with an increase in the elevation angle, as shown in Figure 4.

This effect becomes larger at higher elevations, causing partial observation of neighboring discretized grids at high-elevation angles. The relationship between the effective azimuth viewing angle $\gamma_{eff}$ with elevation angle θ is given by Equation (Equation 6).
(6)$$\gamma_{eff}=\arccos\left[\frac{\cos^{2}\theta -1+\cos2^{{}^{\circ}}}{\cos^{2}\theta }\right]$$

Instantaneous Cartesian position measurements for these RSOs are then generated and corrupted by white noise to model imperfect measurements.
(7)$$y_{k}=h\left(x_{k}\right)+v$$
where *h* is the observation function that outputs the Cartesian coordinates of the RSO and v is a zero mean Gaussian white noise with covariance R:v∼N(0,R), where
$$R=\left[\begin{array}{ccc} 10^{6} & 0 & 0\\ 0 & 10^{6} & 0\\ 0 & 0 & 10^{6} \end{array}\right]$$

### 2.6. Kalman Filter Update

For each observed RSO, a UKF update on the RSOs’ states and covariance is then carried out using the noisy measurement. The UKF uses the unscented transformation to select a minimal set of sigma points around the mean value to represent the RSO states and their associated uncertainties. The following parameters for the unscented transform is used: α=0.001, β=2, and κ=0. The values α and κ influence how far the sigma points are away from the mean. The values were chosen based on typical values used for UKF [22]. The sigma points are then transformed using the same observation function h(·) in Equation (Equation 7). The empirical mean and covariance of the transformed sigma points are then calculated and used to update the RSO’s estimated state and covariance. The readers are referred to the book chapter by Wan and Van Der Merwe (2001) [23] for a complete discussion and implementation details of the UKF.

## 3. Deep Reinforcement Learning Agent

The SSA sensor tasking problem can be formulated as a Partially Observable Markov Decision Process (POMDP) where the full environment state is not presented to the DRL agent during the decision-making process. A POMDP is defined as a tuple {S,A,R,T,Z,O,γ} where S is the set of partially observable states describing the environment, A is the set of actions that an agent may take, R:S×A→R is the reward function that maps the state and action to a reward, $T\left(s^{\prime },s,a\right)=p\left(s^{\prime }|s,a\right)$ is the set of state transition functions that map a state-action pair onto a distribution of state at the next time step, Z is the set of observations that the agent can receive about the environment, $O\left(z^{\prime },s,a\right)=p\left(z^{\prime }|s,a\right)$ is the set of observation probabilities, γ∈[0,1) is a discount factor. The goal of the POMDP is to formulate an action policy that maximizes the expected cumulative discounted future reward given by
(8)$$\max_{a_{t}}E\left[\sum_{t=t_{0}}^{\infty }\gamma^{t-t_{0}}R\left(s_{t},a_{t}\right)\right]$$

At each time step, the SSA sensor tasking environment is in some unobserved environment state s∈S, where the environment state s consists of the state and covariance of all RSOs within the environment. However, the environment state s is not fully observable; the agent can only observe the state of a partial set of the RSOs currently in the sensor’s field of regard and partially observe the covariance of these RSOs. The observable states are governed by the set of observation probabilities $O(z^{\prime },s,a)$. Specifically, the observation z∈Z consists of the 90 × 19 × 11 observation grid outlined in Section 2.2. Based on the observation z, the agent then selects a pointing direction a∈A which in turn causes the environment to transition to state $s^{\prime }\in S$ with probability $T(s^{\prime },s,a)$, i.e., the RSOs’ state and covariance are propagated based on the action time and the new pointing direction a could result in successful detection of RSO and a corresponding decrease in the uncertainties associated with those RSOs. Finally, the agent receives a new observation $z^{\prime }$ and an instantaneous reward R(s,a). Then, the process is repeated until a termination criterion is reached.

The computational complexity of classical POMDP value iteration methods are exponential in the action and observation space. The SSA sensor tasking problem for a single sensor is still highly intractable even after the environment discretization due to the large discrete action space of 1710 possible pointing directions and the continuous observation space of 90×19×11. Hence, DRL is leveraged in this work to solve the SSA sensor tasking problem. Various deep neural network architectures consisting of fully connected layers (FCL) and convolution neural network (CNN) layers are trained in this work to solve the SSA sensor tasking problem. FCL consists of groups of neurons that have trainable weights and biases, where neurons between two adjacent layers are fully pairwise connected. Each neuron will perform a dot product on its inputs and sometimes passes its output through a non-linear activation function. The activation function is responsible for introducing non-linearity to the system and subsequently enabling the neural network to approximate arbitrary complex non-linear functions. The fully connected layer can be represented by Equation (Equation 9).
(9)y=f(Wx+b)
where x and y are the input and output vectors, respectively. W and b are trainable parameters corresponding to the weight matrix and bias vector, respectively. *f* is an element-wise activation function. The CNN layer consists of a sliding window that moves along the input matrix, and operation is only carried out on input data located within the current sliding window. A convolution layer can be represented by Equation (Equation 10).
(10)$$y\left(i,j\right)=\sum_{m}\sum_{n}I\left(m,n\right)x\left(i-m,j-n\right)$$
where *I* is the convolution kernel, *m* and *n* are the dimensions of the sliding window. Compared to fully connected layers, where each output unit interacts directly with every input unit, convolution layers allow for sparse interactions or sparse connectivity, where each output unit only interacts directly with a local subset of the input unit. The convolution layers’ sparse connectivity allows for better spatially correlated data extraction from the input data. A summary of the various DRL architectures (*Dense-v1*, *CNN-v1*, *CNN-v2*, and *CNN-v3*) studied in this work is shown in Table 4. The CNN-based models are inspired by the DRL architectures used in past literature in solving the Atari video games [24,25].

In *CNN-v1*, *CNN-v2* and *CNN-v3*, we experimented with varying the “width” of the neural network. An increasingly narrower network is used, where each subsequent model’s kernel size is reduced. The narrower network forces greater compression onto the input data and can help to prevent over-fitting. This is due to the DRL agent being limited to using a smaller set of parameters to formulate the action policy. All DRL agents studied in this work use an actor-critic model, where two almost identical neural networks are set up and optimized concurrently. The only difference between the actor and critic networks is the dimension of the final layer, where the actor network has a final output layer with 1710 states, while the critic network only has a single final output state. The actor network is responsible for generating the action policy, which consists of an action probability distribution. On the other hand, the critic network outputs a value function for the current state, and the value function is used as feedback to improve the action policy. The *Dense-v1* DRL agent consists of a single shared layer, while the CNN-based DRL agents have no shared layer between the actor and critic networks. Figure 5 shows the neural network architecture for the *Dense-v1* DRL agent.

A rectified linear unit (ReLU) activation function is used for all layers except for the final layer, where a soft-max activation function is used. The outputs of the intermediate layers are normalized to a standard normal distribution using layer normalization. The layer normalization helps to stabilize the hidden state dynamics and prevent dead neurons. The final output of the actor model is post-processed using an action elimination layer. The action elimination layer further penalizes non-rewarding actions (actions leading to an empty field of view) and is formulated based on concepts presented in [26]. This encourages the DRL agents to dedicate more time to exploring rewarding action space and speeds up the convergence of the agent.

Tensorflow [27] and the Ray Tune toolbox [28] are used to implement and train the DRL agents. The DRL agents are trained with the in-house SSA environment on a population of 100 randomly initialized GEO RSOs using a combination of proximal policy optimization [29], and population-based training [30]. The training is done using the high-performance computing (HPC) resources provided by the MIT SuperCloud and Lincoln Laboratory Supercomputing Center [31].

Training of DRL agents can be sensitive to the training hyperparameters, whereas non-optimal training hyperparameters can lead to slow learning and sometimes lead to divergence of the DRL agent. Population-based training is used in this work to overcome this issue, where a population of DRL agents and their hyperparameters are jointly optimized [30]. Population-based training was successfully applied to a wide-ranging suite of challenging DRL problems and demonstrated improved training and final performance. Under the population-based training, a population of DRL agents is first initialized and trained with random training hyperparameters. Then, at fixed training intervals, the population of DRL agents is ranked based on a user-defined performance metric, and the neural network and training hyperparameters of the top-performing agents are exploited to replace the worst-performing agents. The training hyperparameters of the replaced agents are then either perturbed or re-sampled from the provided mutation sampling range. The population-based training algorithm is used to optimize the learning rate and entropy coefficient, and a population size of 20 DRL agents is used. The hyperparameters initialization and mutation sampling range are shown in Table 5.

Meanwhile, the other training hyperparameter values are kept constant and are given in Table 6. These training hyperparameter values were set based on the training hyperparameter of the top-performing agent from the author’s previous works [32].

The performance of the DRL agents is susceptible to the reward function, where the reward function needs to accurately and efficiently describe the agent’s objectives. The reward function defined by Equation (Equation 11) that is constructed based on the objective function discussed in Blackman (1986) [33] is used to trained the DRL agents. The reward function promotes actions that result in a large reduction in position uncertainties.
(11)$$rew=\arg\max_{a_{k}}\left[tr\left(P_{k|k-1}^{\left(a_{k}\right)}\right)-tr\left(P_{k|k}^{\left(a_{k}\right)}\right)\right]$$
where $P_{k|k-1}^{\left(a_{k}\right)}$ and $P_{k|k}^{\left(a_{k}\right)}$ are the a priori and a posteriori covariance of RSO $a_{k}$, respectively, and $a_{k}$ is the set of RSOs within the sensor’s field of view. We did not utilize a different final reward function at the end of the rollout or episode.

## 4. Results and Discussion

Two myopic policies are selected as the baseline to benchmark the performance of the DRL agent. These myopic policies only consider the immediate reward from the current time step and are highly computationally efficient. They do not take into account any long-term effects of their current action. The first myopic scheduler uses a greedy policy that picks the RSO with the largest uncertainties within the sensor’s field of regard to be observed regardless of the required slew angles. The action policy for the greedy policy is given by Equation (Equation 12).
(12)$$a^{*}=\arg\max_{a}\left(tr[P_{k|k-1}^{i}]\right)$$
where $tr[P_{k|k-1}^{i}]$ is the trace of a posteriori state covariance for RSO *i* at the time step k−1.

The greedy policy does not consider the required action time when selecting the next action. However, each action has an associated cost where a different amount of action time is needed for slewing due to the finite slew rate. Instead of making a large slew motion to observe the highest uncertainty RSO located on the other end of the night sky, it is sometimes more beneficial to take multiple smaller movements and observe other RSOs within the catalog while moving in the direction of the high uncertainty RSO. The second myopic policy (advanced greedy policy) considers the action cost by discounting the RSO mean uncertainties with the required action time before picking the most rewarding RSO to be observed. The action policy of the advanced greedy policy is given by Equation (Equation 13).
(13)$$a^{*}=\arg\max_{a}\left(tr\left[P_{k|k-1}^{i}\right]\;\cdot \;\delta t_{i}^{-1/m}\right)$$
where $tr[P_{k|k-1}^{i}]$ is the trace of a posteriori state covariance for RSO *i* at the current time step k−1, $\delta t_{i}$ is the action slew time to move from the current pointing direction to observe RSO *i*, and *m* is a constant. 100 Monte Carlo runs are used to identify the optimum value for the constant *m*, where the advanced greedy policy is evaluated with different values of *m* (ranging from 1 to 20) to identify the optimum scaling factor. For an environment with 100 GEO RSOs and an observation window of 90 min, a *m* value of 10 resulted in the best performance among all performance metrics considered.

The DRL agents are evaluated over two different performance metrics; the number of unique RSOs observed and mean RSO uncertainties at the end of the 90-min observation window. Figure 6a,b show the mean RSO covariance and number of unique RSOs observed over a single-seeded rollout, respectively. The DRL agents can outperform both myopic policies in both performance criteria. All DRL agents outperformed the myopic policies on both performance metrics after 3400 training iterations. During the first half of the observation window, the advanced greedy policy could outperform some DRL agents and have a lower mean trace covariance. This is due to its myopic nature, where highly uncertain RSO is selected to be observed at each time step. However, during the second half of the observation window, the DRL agents benefited from their long-term planning capabilities and arrived at a lower final mean trace covariance compared to both myopic policies. The myopic policies did not perform as well on the second performance metric, where less than 70% of the RSOs in the environment are observed by the myopic policies at the end of the observation window. This is mainly due to the myopic policies not being formulated for this performance criterion. Although the DRL agents were not explicitly trained for this performance criterion, they could observe almost all (around 85%) of the RSOs within the SSA environment. However, due to the random initialization of the RSOs, it is impossible to observe all RSOs over the 90-min observation window as a subset of the RSO population never enters the sensor’s field of regard as shown in Figure 2b. Here, it is shown that it is feasible to have a single policy that optimizes multiple performance criteria or operator objectives through the usage of DRL agents. Furthermore, the reward function can also be formulated to incorporate numerous performance criteria.

Figure 7a,b show the pointing direction selected by the *Dense-v1* and *CNN-v1* DRL agents over the same seeded rollout of 90 min. The *Dense-v1* DRL agent shows a more erratic behavior in its pointing direction compared to the *CNN-v1* DRL agent. The *CNN-v1* DRL agent slew in a consistent azimuth direction with no backtracking, and this consistent azimuth slewing motion ensures that there are always previously unobserved RSO within proximity to the current pointing direction. The other CNN-based DRL agents also displayed a similar trend in their selected azimuth-pointing direction. This highlights that the CNN-based DRL agents could exploit spatial information to efficiently map out a scan path without backtracking in its azimuth-pointing direction.

The performance of the DRL agents and myopic policies are then evaluated via 100 Monte Carlo runs. The 100 Monte Carlo runs ensure that a diverse set of scenarios are considered for performance analysis and will provide a better statistical comparison between the various action policies. Each Monte Carlo run uses a differently seeded environment with randomly initialized RSO states and covariance. Figure 8a,b show the aggregate performance of the DRL agents and the myopic policies. The DRL agents outperformed the myopic policies on both performance metrics. The DRL agents have similar performance on the first performance metric. The *CNN-v3* DRL agent that was trained for 3400 training iterations has the worst performance in terms of final mean RSO uncertainties, followed by the FCL-based DRL agent (*Dense-v1*). The *CNN-v2* DRL agent trained for 6100 training iterations has the best aggregate performance for both performance criteria. From the performance of the *CNN-v2* and *CNN-v3* DRL agents, it can be observed that the DRL agents can be further improved with additional training iterations, where there was a noticeable performance gain for the *CNN-v3* DRL agent when the number of training iterations was tripled. On average, the DRL agents could observe 80–96% of the 100 RSOs in the SSA environment, with the FCL-based DRL agent (*Dense-v1*) having the worst performance on the second performance metric. The wider *CNN-v1* and *CNN-v2* DRL agents were able to perform better than the narrower *CNN-v3* DRL agent. The advanced greedy policy performed better than the naive greedy one as it considers action costs during decision-making. Due to the myopic nature of the greedy policy, it tends to make large slewing motions to observe the most uncertain RSO in the field of regard. Thus, a significant portion of the observation window was utilized for slewing motion, resulting in a lower number of measurements and hence the worst performance on both metrics.

### 4.1. Robustness to Changes in RSO Orbital Regime

We then looked into the robustness of the DRL agents to changes in the RSO population, particularly the orbital regime of the RSOs. Optical sensors primarily used for tracking GEO RSOs are also frequently tasked to observe objects in the LEO regime. Therefore, the DRL agents previously trained in an SSA environment with GEO RSOs are applied to an SSA environment consisting of only LEO RSOs. The LEO orbital regime is more challenging mainly due to the “faster moving” RSOs, where the LEO RSOs rapidly enter and exit the field of regard. On the other hand, GEO RSOs tend to be slower moving in the night sky and provide ample observation opportunities throughout the observation window. A new KDE model for the LEO orbital regime is constructed using the methodology outlined in Section 2.1. Figure 9a,b show the instantaneous number of RSOs within the field of regard and the cumulative number of unique RSOs that have entered the sensor’s field of regard since the beginning of the observation window, respectively. Although both KDE models resulted in a similar percentage of unique RSOs that entered the sensor’s field of regard at the end of the observation window, the LEO KDE model has a much lower instantaneous number of RSOs throughout the observation window.

On average, the LEO KDE model has 16 RSOs within the field of regard, whereas the GEO KDE model has an average of 86 RSOs. To compensate for the lower number of instantaneous RSOs in the field of regard, the number of RSOs in the environment is quadrupled from 100 RSOs to 400 RSOs. We also retrained the best performing DRL agent (*CNN-v2*) on an SSA environment with 100 LEO RSOs.

100 Monte Carlo runs are then carried out on the SSA environment with 400 LEO RSOs and the aggregate performance of the DRL agents and myopic policies are shown in Figure 10a,b. The CNN-based DRL agents trained using GEO RSOs could not outperform the advanced greedy policy on the first performance metric (mean trace covariance of all RSOs at the end of the observation window). However, they performed better than all myopic policies on the second performance metric, where they could observe a higher number of unique RSOs at the end of the observation window. The “faster moving” LEO RSOs behave differently than the training data, and the DRL agents had difficulties adapting to the “faster moving” LEO RSOs. On the other hand, the FCL-based DRL agent (*Dense-v1*) could only outperform the naive greedy policy on both performance criteria. The FCL-based DRL agent has lower robustness to changes in the RSO orbital regime than the CNN-based DRL agents. The DRL agent trained using LEO RSOs (*CNN-v2-LEO*) outperformed all other policies on both performance metrics. It was also observed that the performance of the *CNN-v2-LEO* DRL agent improved slightly with more training iterations.

We then evaluated the performance of the *CNN-v2-LEO* DRL agent on the GEO SSA environment. The *CNN-v2-LEO* DRL agent performed similar to the *CNN-v2* DRL agent that was trained on the GEO SSA environment as shown in Figure 11. There is a slight degradation in performance for the *CNN-v2-LEO* DRL agent at the expanse of increased robustness toward variation in the RSO orbital regime. DRL agents trained on the more challenging LEO SSA environment could generalize better and perform well across different orbital regimes. Furthermore, as the *CNN-v2-LEO* DRL agent was trained using 100 LEO RSOs, there is a lower instantaneous number of RSOs within the field of regard which further increase the complexity of the problem due to the sparser reward.

### 4.2. Robustness to Changes in the Length of Observation Window

The observation window for ground-based optical telescopes can vary depending on the weather conditions and the local sunrise and sunset time. Thus, there is a need to analyze the robustness of the DRL agents toward variation in the observation window. The observation window length is gradually increased from the initial 90 min to 8 h to simulate better the variation of real-life observation windows based on the astronomical sunset to sunrise time due to seasonal changes. The DRL agents were not retrained for the longer observation window. As the GEO KDE model is observation window dependent, a new GEO KDE model must be constructed for the different observation window lengths.

Figure 12 shows the statistics of the RSOs within the field of regard for different KDE models over an 8 h observation window and highlights the need for a different KDE model for the various observation window.

For example, when the GEO 90 min KDE model is used for an 8 h propagation, we notice that the SSA environment starts with most of the RSOs (86%) already within the field of regard. However, after 90 min, the number of instantaneous RSOs in the field of regard rapidly drops, where less than one-third of the initial RSOs remain after propagating for 330 min. On the other hand, only 65% of the 100 RSOs in the environment enter the sensor’s field of regard when the 480 min KDE model is used for an observation window of 90 min. Similar trends are observed when there is a mismatch between the KDE model used and the observation window length. Ideally, we would prefer that the number of RSOs within the field of regard to stay relatively consistent and for more than 95% of the RSO population to have entered the field of regard by the end of the observation window.

The aggregate performance of the DRL agents and myopic policies under varying observation window lengths are shown in Figure 13. The performance of all action policies gradually increases as the length of the observation window increases. This is mainly due to the extended observation windows providing additional time for the various policies to observe more RSOs, thus generally improving their performance.

The DRL agent observed a more significant performance gain than the myopic policies for the first performance metric. As time passes, the advanced greedy policies start selecting actions that make large changes in pointing direction to observe the highly uncertain RSOs that have just entered the field of regard.

### 4.3. Robustness to Changes in Observer Location

To ensure continuous complete coverage of Earth orbit, the SSA sensors working collaboratively must be deployed evenly across the Earth. Hence, it is crucial to analyze the sensitivity of the DRL agents to variations in the observer location. The DRL agents that are trained using a sensor located in Minnesota ($44.9778^{{}^{\circ}}$ N, $93.2650^{{}^{\circ}}$ W) are applied to the tasking of sensors located in Miami ($25.7330^{{}^{\circ}}$ N, $80.1650^{{}^{\circ}}$ W) and Australia ($21.8171^{{}^{\circ}}$ S, $114.1666^{{}^{\circ}}$ E). The sensor parameters, such as field of view, minimum viewing horizon, and slew rate, are kept constant. Figure 14 shows the aggregate performance of the various policies at the different sensor locations.

A similar trend was observed across all three sensor locations, where the DRL agents outperformed all myopic policies. The CNN-based *CNN-v2* DRL agent has the best performance on both performance metrics, followed by the FCL-based *Dense-v2* DRL agent, the advanced greedy policy, and finally the naive greedy policy. There is a slight degradation in the performance of the DRL agents when applied to the new sensor locations. The performance of the DRL agents can be further improved by utilizing transfer learning and training them further with the new sensor location. The trained DRL agents are robust to variation in the sensor location and can be deployed to any arbitrary location without any significant modifications. The observation formulation used in the SSA environment transforms the RSO data into a location-agnostic observation space, where the observation array is always centered around the current pointing direction.

### 4.4. Robustness to Changes in Sensor Slew Rate

The slew rate of ground-based sensors can vary greatly depending on the actuator, size, and mass. Hence, it is helpful to analyze the robustness of the DRL agent to variation in the sensor slew rate. During the training setup, the DRL agents were trained using the sensor parameters of an amateur-grade telescope with a slew rate of 4.55 s for every 4${}^{{}^{\circ}}$. The DRL agents were applied to a sensor capable of slewing twice as fast at a rate of 2.5 s for every 4${}^{{}^{\circ}}$, while the other sensor parameters and location are kept the same. The observation grid is reformulated by considering the new slew rate, where a lower propagation time is used for each partition to reflect the lower action slew time. The new observation grid is formed using RSO states from the following future time steps: 13.25 s, 29 s, 45.25 s, 61.5 s, 77.75 s, 94 s, and 110.25 s. Figure 15 shows the aggregate performance of the DRL agents using 100 Monte Carlo runs with a faster slew rate of 2.5 s for every 4${}^{{}^{\circ}}$. The DRL agents outperformed the myopic policies for both performance criteria. All DRL agents performed similarly in terms of final mean trace covariance at the end of the observation window. However, the CNN-based agents could observe a slightly higher number of unique RSOs than the FCL-based DRL agent. Due to the faster slew rate, the DRL agents could observe more unique RSOs compared to the previous scenarios. On average, the DRL agents could observe more than 90% of the 100 GEO RSOs within the SSA environment. The DRL agents have been shown to be robust to changes in sensor slew rate. Our formulation of the partitioned observation grid increases the robustness of the DRL agents to variation in sensor parameters by pre-propagating the RSOs and accurately reflecting what the DRL agent is expected to observe.

## 5. Conclusions

Modern society relies heavily upon space infrastructures for day-to-day operations, ranging from communication, guidance and navigation, and weather forecasts to space imagery. These space infrastructures are susceptible to collisions with debris, derelict space objects, and other active satellites. Moreover, the number of conjunction events has steadily increased in recent decades with the increasing space activities. Hence, achieving space situational awareness (SSA) in an accurate and timely fashion is of utmost importance. Better sensor management algorithms are thus required to efficiently allocate our sensing resources to safeguard our space assets and ensure sustainable space usage.

In this work, deep reinforcement learning (DRL) methods are leveraged to overcome the curse of dimensionality inherent in the space situational awareness sensor tasking problem. An in-house SSA environment is developed to train and evaluate the performance of the DRL agents efficiently. The SSA environment can support arbitrary sensor location, various RSOs, observation windows, and sensor properties (action slew time, settle time, dwell time, measurement model, measurement noise, and process noise). Four DRL agents are trained using population-based training and proximal policy optimization for the SSA sensor tasking problem. Both fully connected and convolution neural network DRL agents are explored in this work. The DRL agents outperformed myopic policies and achieved lower mean RSO uncertainties and a higher cumulative number of unique RSOs observed over a 90-min observation window. The DRL agents’ robustness to changes in orbital regimes, observation window length, observer location, and sensor’s slew rate are also studied. The DRL agents have shown robustness to most of these variations and continue outperforming the myopic policies. The robustness of the DRL agents allows them to be applied to any arbitrary sensors and scenarios.

Some of our proposed future works include extending the current SSA environment to support multiple sensors working together concurrently and improving the fidelity and realism of the SSA environment by incorporating detection probability due to weather and solar illumination effects. Furthermore, more advanced neural network architectures will be explored.

## Figures and Tables

**Figure 1 sensors-22-07847-f001:**
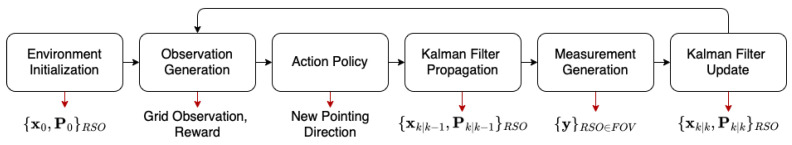
Process flow for the space situational awareness environment.

**Figure 2 sensors-22-07847-f002:**
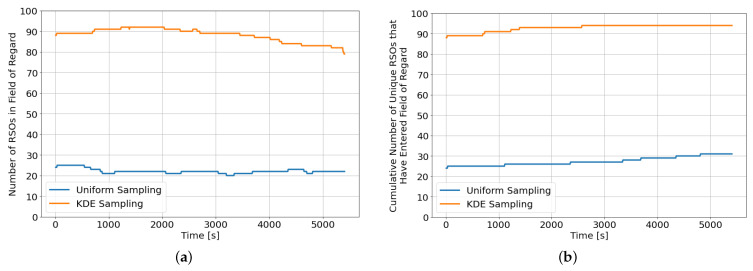
Comparisons between KDE sampling and uniform sampling over a single rollout of 90 min. (**a**) Number of RSOs in field of regard; (**b**) Cumulative number of unique RSOs in field of regard since the beginning of the episode.

**Figure 3 sensors-22-07847-f003:**

Discretized observation space and RSO propagation time used to account for varying action slew time.

**Figure 4 sensors-22-07847-f004:**
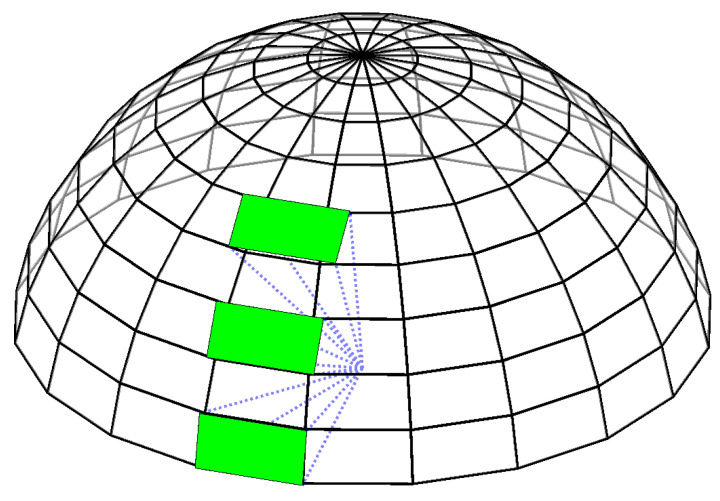
Discretization of the field of regard. The green patches correspond to the regions in the field of regard observed by the sensor at different elevations.

**Figure 5 sensors-22-07847-f005:**
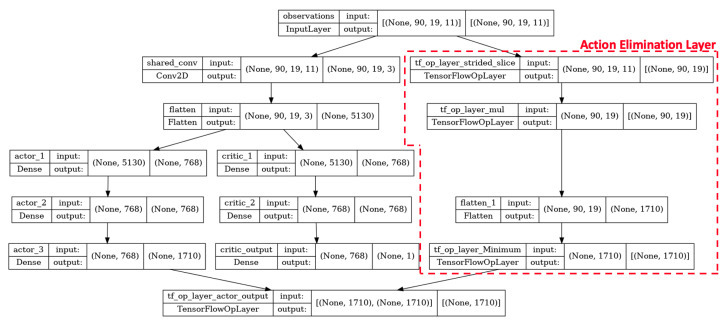
Neural network architecture for *Dense-v1* deep reinforcement learning agent.

**Figure 6 sensors-22-07847-f006:**
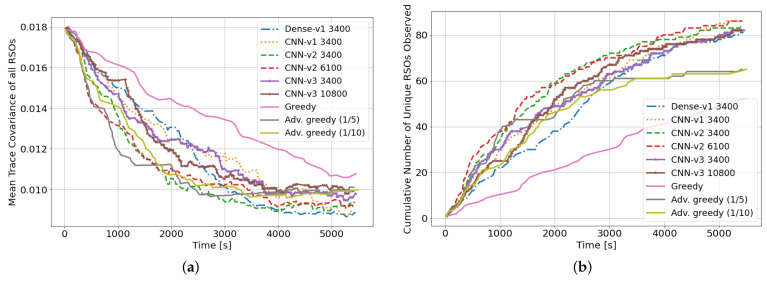
Performance of DRL agents with different architectures trained using 100 GEO RSOs, evaluated on a single-seeded rollout of 90 min. (**a**) Mean trace covariance of all RSOs for one rollout episode (lower is better); (**b**) Cumulative number of unique RSOs observed for one rollout episode (higher is better). Note: *CNN-v1 3400* represents a *CNN-v1* DRL agent trained for 3400 training iterations.

**Figure 7 sensors-22-07847-f007:**
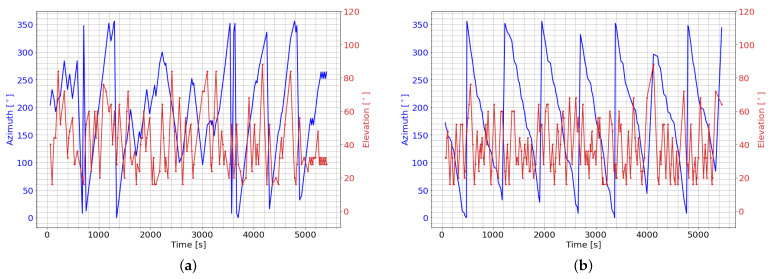
Pointing directions selected by DRL agent. (**a**) Azimuth and elevation pointing direction selected by the *Dense-v1 3400* DRL agent; (**b**) Azimuth and elevation pointing direction selected by the *CNN-v1 3400* DRL agent.

**Figure 8 sensors-22-07847-f008:**
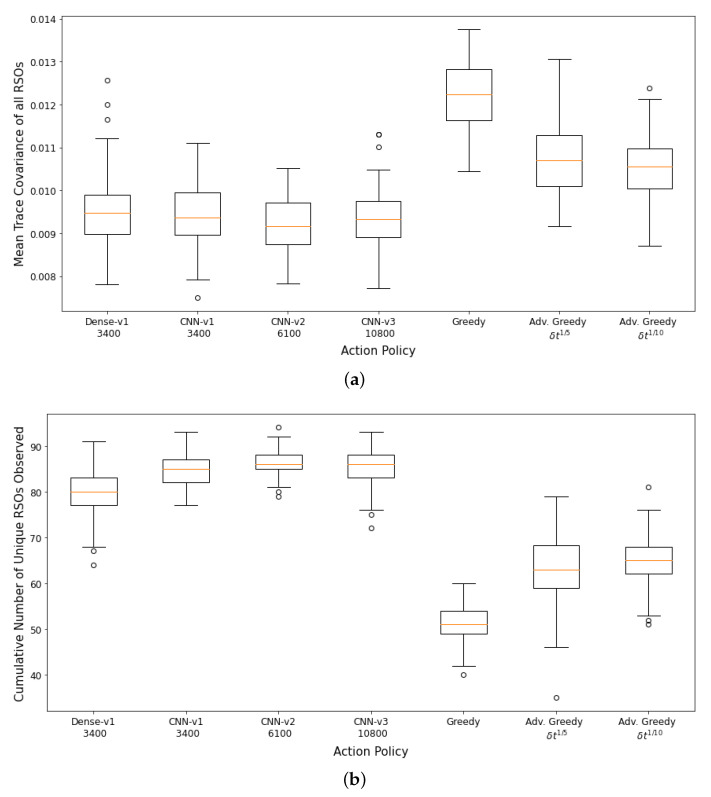
Aggregate performance of the DRL agents with 100 randomly initialized GEO RSOs, evaluated over 100 seeded rollouts of 90 min each. (**a**) Distribution of the final mean trace covariance at the end of the observation window (lower is better); (**b**) Distribution of the cumulative number of unique RSOs observed at the end of the observation window (higher is better). Note: *CNN-v1 3400* represents a *CNN-v1* DRL agent trained for 3400 training iterations.

**Figure 9 sensors-22-07847-f009:**
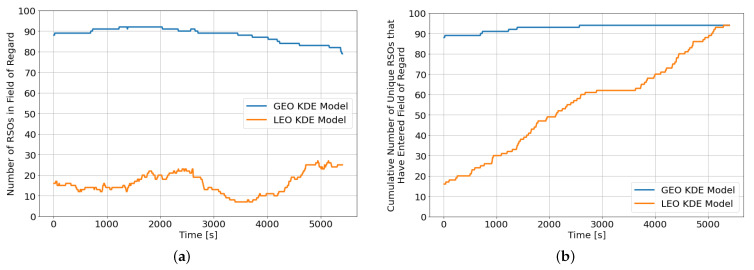
RSO distribution for population initialized for different orbital regimes. (**a**) Number of RSOs in field of regard; (**b**) Cumulative number of unique RSOs in field of regard since the beginning of the episode.

**Figure 10 sensors-22-07847-f010:**
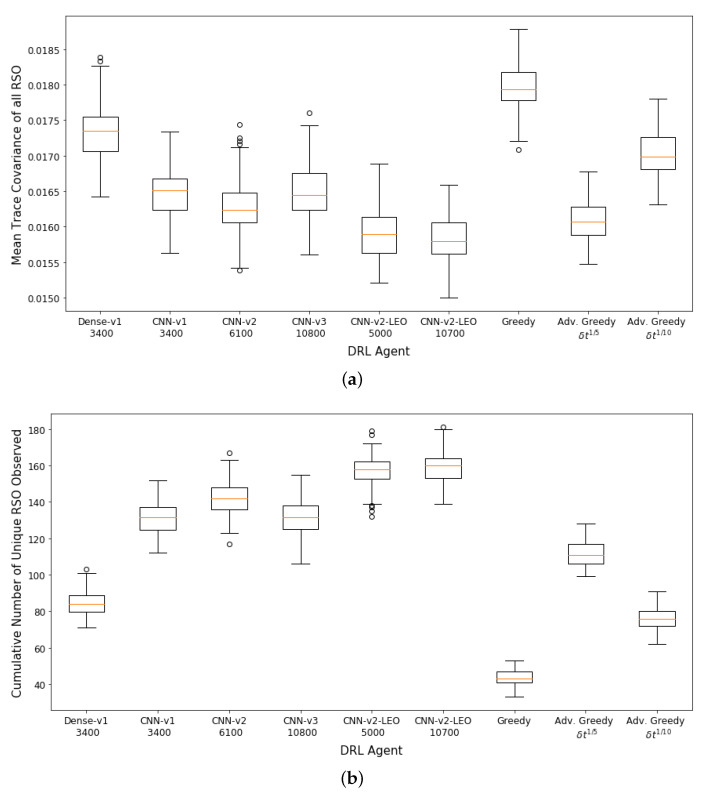
Aggregate performance of the DRL agents with 400 randomly initialized LEO RSOs, evaluated over 100 seeded rollouts of 90 min each. (**a**) Distribution of the final mean trace covariance at the end of the observation window (lower is better); (**b**) Distribution of the cumulative number of unique RSOs observed at the end of the observation window (higher is better). Note: *CNN-v1 3400* represents a *CNN-v1* DRL agent trained for 3400 training iterations.

**Figure 11 sensors-22-07847-f011:**
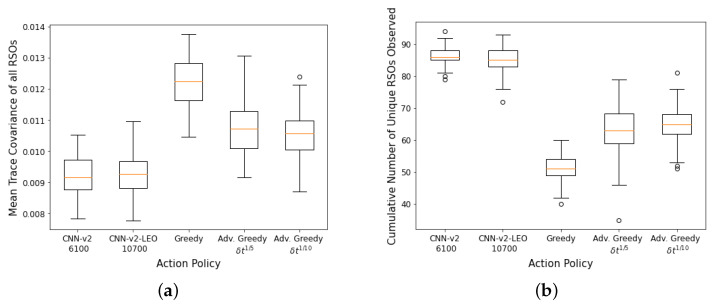
Aggregate performance of selected DRL agents with 100 randomly initialized GEO RSOs, evaluated over 100 seeded rollouts of 90 min each. (**a**) Distribution of the final mean trace covariance at the end of the observation window (lower is better); (**b**) Distribution of the cumulative number of unique RSOs observed at the end of the observation window (higher is better). Note: *CNN-v1 3400* represents a *CNN-v1* DRL agent trained for 3400 training iterations.

**Figure 12 sensors-22-07847-f012:**
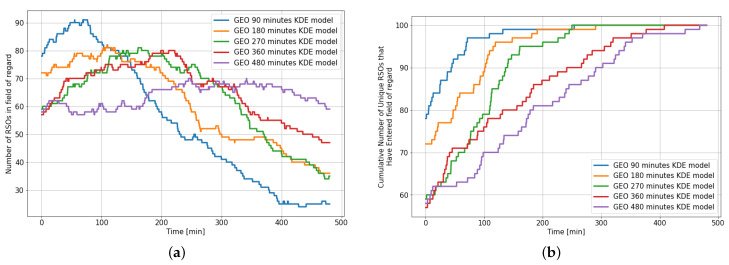
Statistics of RSOs within the field of regard over an 8 h observation window for 100 RSOs using different GEO KDE models. (**a**) Number of RSOs within the field of regard; (**b**) Cumulative number of unique RSOs within the field of regard.

**Figure 13 sensors-22-07847-f013:**
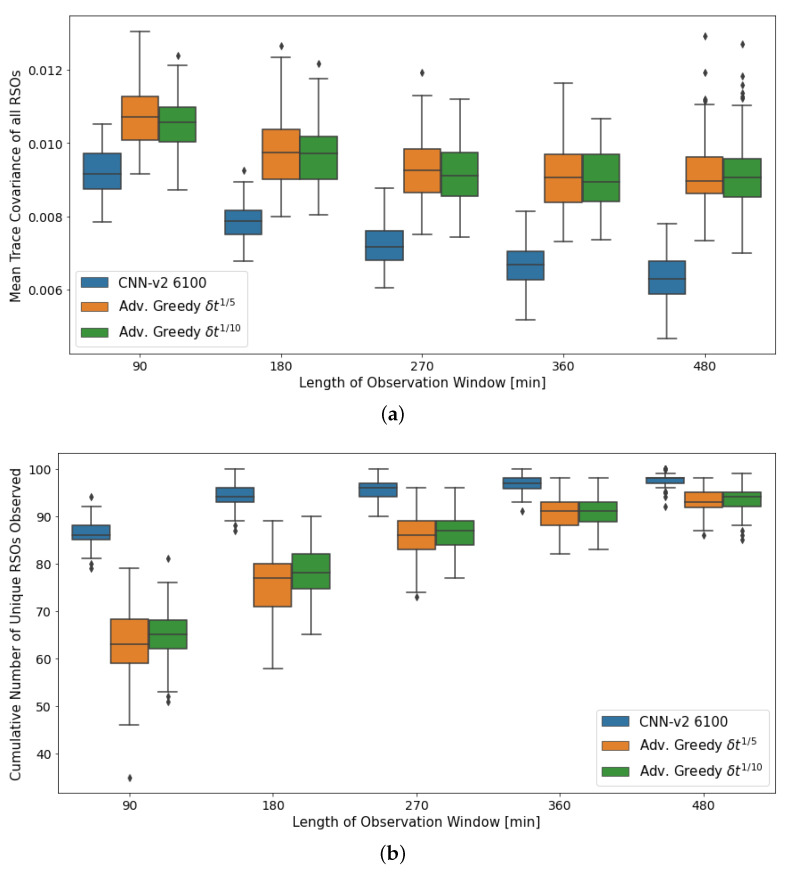
Aggregate performance of the DRL agents with 100 randomly initialized GEO RSOs, evaluated over 100 seeded rollouts of varying observation window length. (**a**) Distribution of the final mean trace covariance at the end of the observation window (lower is better); (**b**) Distribution of the cumulative number of unique RSOs observed at the end of the observation window (higher is better). Note: *CNN-v2 6100* represents a *CNN-v2* DRL agent trained for 6100 training iterations.

**Figure 14 sensors-22-07847-f014:**
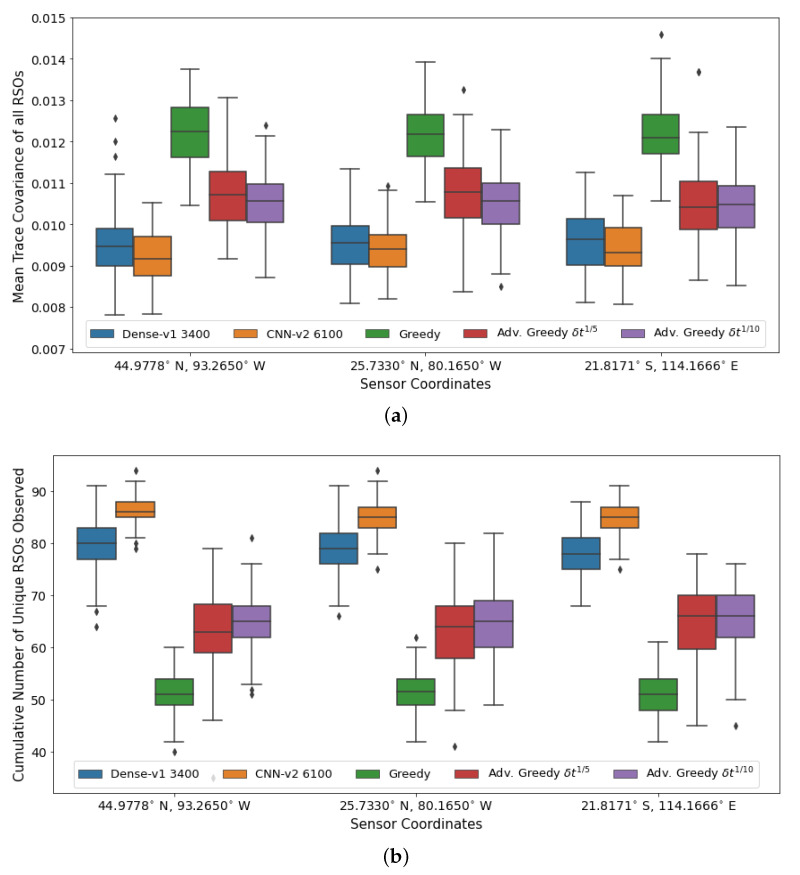
Aggregate performance of the DRL agents with 100 randomly initialized GEO RSOs, evaluated over 100 seeded rollouts of 90 min each and using varying sensor coordinates. (**a**) Distribution of the final mean trace covariance at the end of the observation window (lower is better); (**b**) Distribution of the cumulative number of unique RSOs observed at the end of the observation window (higher is better). Note: *CNN-v2 6100* represents a *CNN-v2* DRL agent trained for 6100 training iterations.

**Figure 15 sensors-22-07847-f015:**
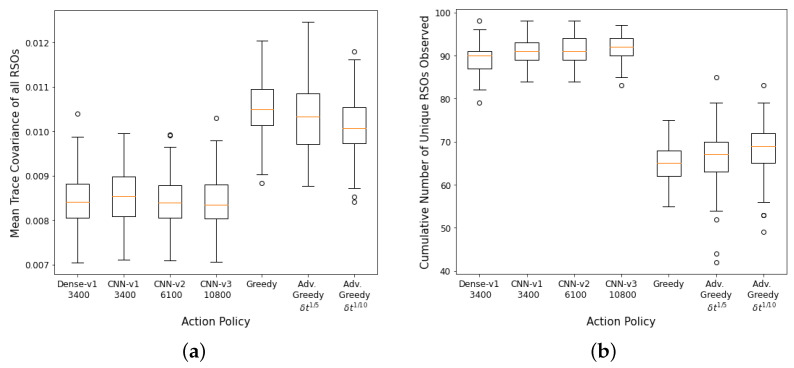
Aggregate performance of the DRL agents with 100 randomly initialized GEO RSOs, evaluated over 100 seeded rollouts of 90 min each and using a ground-based sensor with a faster slew rate of 2.5 s for every 4${}^{{}^{\circ}}$. (**a**) Distribution of the final mean trace covariance at the end of the observation window (lower is better); (**b**) Distribution of the cumulative number of unique RSOs observed at the end of the observation window (higher is better). Note: *CNN-v1 3400* represents a *CNN-v1* DRL agent trained for 3400 training iterations.

**Table 1 sensors-22-07847-t001:** Parameters of Optical Sensor.

Sensor Parameter	Value
Field of View	$4^{{}^{\circ}}\times 4^{{}^{\circ}}$
Minimum Viewing Horizon	$14^{{}^{\circ}}$
Action Slew Rate	7.7 s/first 4${}^{{}^{\circ}}$
	4.55 s/subsequent 4${}^{{}^{\circ}}$
Exposure time	1.3 s

**Table 2 sensors-22-07847-t002:** Sampling bound for diagonal of RSO covariance.

Orbital Element	TLE Covariance from Osweiler (2006) [20]		Sampling Bounds for RSO Initialization	
	First Quartile	Third Quartile	Lower Bounds	Upper Bounds
Inclination, deg${}^{2}$	1.6 $\times \;10^{-8}$	6.9 $\times \;10^{-8}$	1.6 $\times \;10^{-8}$	6.9 $\times \;10^{-8}$
RAAN, deg${}^{2}$	1.9 $\times \;10^{-8}$	1.9 $\times \;10^{-7}$	1.9 $\times \;10^{-8}$	1.9 $\times \;10^{-7}$
Eccentricity	2.3 $\times \;10^{-14}$	5.0 $\times \;10^{-13}$	2.0 $\times \;10^{-14}$	5.0 $\times \;10^{-13}$
Argument of perigee, deg${}^{2}$	3.0 $\times \;10^{-4}$	2.5 $\times \;10^{-2}$	3.0 $\times \;10^{-4}$	2.5 $\times \;10^{-2}$
Mean anomaly, deg${}^{2}$	3.6 $\times \;10^{-4}$	2.6 $\times \;10^{-2}$	3.6 $\times \;10^{-4}$	2.6 $\times \;10^{-2}$
Mean motion, (rev/day)${}^{2}$	2.0 $\times \;10^{-10}$	4.0 $\times \;10^{-8}$	2.0 $\times \;10^{-8}$	4.0 $\times \;10^{-6}$

**Table 3 sensors-22-07847-t003:** Observation information for each grid in the observation.

Layer	Data
1	Number of RSOs in grid
2	Elevation angle of RSO in grid
3	Azimuth angle of RSO in grid
4	Range of RSO in grid
5	Rate of change in elevation of RSO in grid
6	Rate of change in azimuth of RSO in grid
7	Rate of change in range of RSO in grid
8	Max of RSO’s trace of covariance in grid
9	Sum of RSO’s trace of covariance in grid
10	Mean of RSO’s trace of covariance in grid
11	Current pointing direction (boolean)

**Table 4 sensors-22-07847-t004:** Deep reinforcement learning neural network architectures.

Architecture	Input Size	Output Size	Layers	Number of Parameters
Mnih et al.	(84×84×4)	4∼18	Conv2d(16,8,4),	1,004,852∼1,008,450
(2013) ^1^ [24]			Conv2d(32,4,2),	
			FCL(256),	
			FCL(Output Size)	
Mnih et al.	(84×84×4)	4∼18	Conv2d(32,8,4),	4,045,476∼4,052,658
(2015) ^1^ [25]			Conv2d(64,4,2),	
			Conv2d(64,3,1),	
			FCL(512),	
			FCL(Output Size)	
Dense-v1	(90×19×12)	1710	Conv2d(3,1,1),	5,846,229
			FCL(768),	
			FCL(768),	
			FCL(Output Size)	
CNN-v1	(90×19×12)	1710	Conv2d(48,8,4),	9,560,190
			Conv2d(80,4,2),	
			Conv2d(80,3,1),	
			FCL(2048),	
			FCL(Output Size)	
CNN-v2	(90×19×12)	1710	Conv2d(32,8,4),	4,207,438
			Conv2d(64,4,2),	
			Conv2d(64,3,1),	
			FCL(1024),	
			FCL(Output Size)	
CNN-v3	(90×19×12)	1710	Conv2d(16,8,4),	2,035,586
			Conv2d(32,4,2),	
			Conv2d(32,3,1),	
			FCL(700),	
			FCL(Output Size)	

^1^ These two models were not used in this work but are provided for reference. *FCL*(*256*) corresponds to a fully connected layer with 256 output nodes, whereas *Conv2d*(*16,8,4*) corresponds to a convolution neural network layer using a kernel with 16 filters and a sliding window of 8 × 8 with a 4 × 4 stride.

**Table 5 sensors-22-07847-t005:** Initialization and mutation range of hyperparameters for hyperparameter optimization with population-based training.

Hyperparameter	Initialization	Mutation Sampling Range
Entropy coefficient, β	[1 $\times \;10^{-2}$, 1 $\times \;10^{-4}$]	[0.0, 1 $\times \;10^{-2}$, 5 $\times \;10^{-3}$, 1 $\times \;10^{-3}$, 5 $\times \;10^{-4}$, 1 $\times \;10^{-4}$]
Learning rate	[5 $\times \;10^{-4}$, 5 $\times \;10^{-6}$]	Uniform(0.0, 1 $\times \;10^{-3}$)

**Table 6 sensors-22-07847-t006:** Hyperparameters used for training.

Training Hyperparameter	Value
Lambda, λ	0.7389
Vf loss coefficient, $c_{1}$	0.6247
Entropy coefficient, β	0.3498
Clip parameter, ϵ	3.7579 $\times \;10^{-4}$
Learning rate	5 $\times \;10^{-4}$
Number of training epoch	1
Minibatch size	273
Training batch size	3472

## Data Availability

Not applicable.

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
