# Peer review of "Optimal Tasking of Ground-Based Sensors for Space Situational Awareness Using Deep Reinforcement Learning"

_sensors, 2022, doi:10.3390/s22207847_

Round 1

Reviewer 1 Report

Summary

This paper studies a real-world problem of high impact, tasking of ground-based sensors for space situational awareness. The goal is to control a single ground-based optical sensor to make its observation more certain and efficient. The input to the control agent is observation grid and the output is a pointing direction consisting of an azimuth angle and an elevation angle. The authors first develop a simulation environment (section2). The deep reinforcement learning (DRL) algorithm is successfully applied to solve the problem. To make DRL work, several training tricks and models are also studied. The reviewer thinks the problem is interesting and the paper fits the scope of the Sensors journal. However, the reviewer has a major concern whether this paper provides significant contribution beyond the conference paper [2].

Questions and Comments

1. As the paper discussed, the SSA sensor tasking problem is a POMDP. However, partial observability usually hinders the success of DRL algorithms. To make DRL work on POMDP, we usually need to apply some approximation or reduce the observability (e.g., concatenating past states). This part seems to be missing, and the reviewer wonder how could we guarantee the optimality of DRL in this case?

2. Since the problem is POMDP, can we try POMDP planning as a compared method? Since there’re only one sensor, the dimensionality doesn’t seem to be an issue.

3. This work is evolved from [1] and [2]. It seems this paper has many same contents (writings and figures) with [2]. What’s more, the reviewer couldn’t tell the difference and major contributions compared to [2]. I would suggest more contents on

(1) Does many of the key ideas in this appear in the previous conference papers?

(2) Does this paper use a significant amount of text, results, data, or figures from the previous conference paper?

I’d also suggest discussing the key contributions of this paper. Are there any other papers that use DRL for this problem?

4. This paper discretize the continuous action space into 19 discrete elevation pointing angles and 90 discrete azimuth pointing angles. The reviewer wonders

(1) Since the action is discrete, why not use value-based methods such as D3QN? We know DQN is usually more suitable for discrete action space. The reviewer would like to know the reason for choosing PPO.

(2) Did the author try the continuous action space directly?

[1] Thomas G. Roberts, Peng Mun Siew, Daniel Jang, and Richard Linares, "A Deep Reinforcement Learning Application to Space-based Sensor Tasking for Space Situational Awareness." (2021).

[2] Siew, Peng Mun, Daniel Jang, and Richard Linares. "Sensor Tasking for Space Situational Awareness Using Deep Reinforcement Learning." AIAA/AAS Astrodynamics Specialist Conference, Big Sky, MT. 2021.

Reviewer 2 Report

In the paper an optimal approach to space situational awareness (SSA) is addressed from the viewpoint of machine learning to obtain optimal solution. In the ininitial part of the paper the debris problem is described, placing it as a hazardous condition on operation of space missions. The topic is definitely up to date, as per e.e generation of debris by Russian anti-satellite defense system, or numerous cases when debris caused threat to space missions. 

Similar reasoning can be found in 10.1088/1538-3873/ab9cc5, to monitor NEO objects. The SSA problem is posed as the optimization problem with constraints. Classical approaches are cited and references, as well as the more modern ones in application to sensor tasking. Eventually, the authors end up on deep reinforcement learning approach. 

Prior to discussion about the paper structure, the authors should give a frank description of both the novelty, as well as the contribution, to allow the readers to easily identify it. 

Figure 1 presents the algorithm behind SSA environment. The question is whether the availability of the model of the optical sensor could be used instead of KFs to improve the quality of observation? Would the solution benefit from any information concerning the model, i.e. a digital twin. 

Section 2.1 - why is the Gaussian PDF assumed? Is this because of lack of crisp picture of NEOs or farther objects, or due to the assumption concerning noise properties. What if any other distribution is assumed? 

The covariance matrix R notation is strange, please turn 'e' into 10^{} notation. 

No argument is given below max in (8) thus it is not possible to understand what the decision variables in this optimization task are. The same holds for the remaining notation. 

Support the use of specific parameter values in Table 6, please. 

Line 399 - in prior tables you separated thousands from hundreds using a comma. The same notation shoudl be used throught the paper. 

The statistical comparison presented in Figure 10 is of importance, as well as the other ones presenting meand and standard values. This is a strong point in the paper. 

To sum up, the paper presents a deeply analyzed approach, well-compared to the other approaches, with in-depth discussion leaving actually no room for questions. I suggest minor revision due to notation, as the discussion in the paper is complete. 

Round 2

Reviewer 1 Report

My previous concerns have been properly addressed or explained in

detail. I believe this paper could bring intersting ideas to the

community. So review decision is acceptance.